# Slow Adaptive Response of Budding Yeast Cells to Stable Conditions of Continuous Culture Can Occur without Genome Modifications

**DOI:** 10.3390/genes11121419

**Published:** 2020-11-27

**Authors:** Joanna Klim, Urszula Zielenkiewicz, Anna Kurlandzka, Szymon Kaczanowski, Marek Skoneczny

**Affiliations:** 1Department of Microbial Biochemistry, Institute of Biochemistry and Biophysics, Polish Academy of Sciences, Pawińskiego 5a, 02-106 Warsaw, Poland; klim@ibb.waw.pl (J.K.); ulazet@ibb.waw.pl (U.Z.); 2Department of Genetics, Institute of Biochemistry and Biophysics, Polish Academy of Sciences, Pawińskiego 5a, 02-106 Warsaw, Poland; ania218@ibb.waw.pl; 3Department of Bioinformatics, Institute of Biochemistry and Biophysics, Polish Academy of Sciences, Pawińskiego 5a, 02-106 Warsaw, Poland; szymon@ibb.waw.pl

**Keywords:** transcriptome, stress alertness, *Saccharomyces cerevisiae*, epigenetic, adaptation, transcription factor

## Abstract

Continuous cultures assure the invariability of environmental conditions and the metabolic state of cultured microorganisms, whereas batch-cultured cells undergo constant changes in nutrients availability. For that reason, continuous culture is sometimes employed in the whole transcriptome, whole proteome, or whole metabolome studies. However, the typical method for establishing uniform growth of a cell population, i.e., by limited chemostat, results in the enrichment of the cell population gene pool with mutations adaptive for starvation conditions. These adaptive changes can skew the results of large-scale studies. It is commonly assumed that these adaptations reflect changes in the genome, and this assumption has been confirmed experimentally in rare cases. Here we show that in a population of budding yeast cells grown for over 200 generations in continuous culture in non-limiting minimal medium and therefore not subject to selection pressure, remodeling of transcriptome occurs, but not as a result of the accumulation of adaptive mutations. The observed changes indicate a shift in the metabolic balance towards catabolism, a decrease in ribosome biogenesis, a decrease in general stress alertness, reorganization of the cell wall, and transactions occurring at the cell periphery. These adaptive changes signify the acquisition of a new lifestyle in a stable nonstressful environment. The absence of underlying adaptive mutations suggests these changes may be regulated by another mechanism.

## 1. Introduction

To obtain microbial biomass for downstream applications, microorganisms are grown either in batch culture or in continuous culture, usually in a chemostat. Batch culture is easier to implement and less costly and requires simpler instrumentation, and, in general, the time necessary to obtain a sufficient quantity of biomass is determined only by the microorganism cell doubling time. On the other hand, continuous culture is lengthier and more costly and requires dedicated instrumentation, yet allows greater control of the growth conditions, long-term stability of cultured cell population physiological and metabolic parameters, and run-to-run reproducibility of these parameters [1]. By minimizing the experimental noise and the number of variables it is easier to expose those important for the given study. These preponderances become particularly advantageous when whole transcriptome studies are performed [2], for which it is important to discern the anticipated transcriptome changes from those pertaining to growth conditions and/or other factors escaping experimenter control. In batch culture, cells experience constant changes in nutrient availability as well as the build-up of byproducts and final metabolites. This is also true for the cells of budding yeast *Saccharomyces cerevisiae*, the model organism in the studies described in this paper. When grown in a medium with glucose as a carbon source its predominant way of obtaining energy is fermentation, which leads to the accumulation of ethanol. The regulatory mechanism called glucose repression prevents budding yeast cells from entering into respiratory metabolism despite the availability of oxygen until glucose is exhausted [3]. At this point, budding yeast cells undergo a diauxic shift from fermentation to oxidation of ethanol, which reprograms the whole metabolism and affects the expression of hundreds of genes [4,5]. This is an extreme example; however, growth phase-specific changes in the transcriptome constitute a common source of artifacts often distorting the resulting data [2] and leading the experimenters astray.

Unquestionably, continuous culture also has limitations [6] and cultured cells do not preserve constant features indefinitely. Indeed, prolonged continuous cultures, typically in limited chemostats and exposed to some selection pressure are commonly employed when microevolution experiments are performed. In these approaches, gradual adaptive changes in cell populations in response to that selection are expected and desired [7,8] and cell cultures are maintained for hundreds or even thousands of generations. On the other hand, in typical transcriptome studies, continuous culture should not be carried out for too long to avoid admixture of adaptive-borne changes. To prevent this, cells should be harvested soon after a culture reaches the steady-state and the number of generations in the continuous culture should not exceed twenty [7].

The phenomenon of adaptation in a prolonged culture is well known. It has been demonstrated in numerous microevolution studies that during the adaptive phase of continuous culture, mutations accumulate in the cell population gene pool, and cells gaining fitness due to these mutations dominate the population [9,10,11,12]. Adaptive changes in gene expression within the cell population have also been observed in whole-transcriptome studies [7,13,14,15]. It seems obvious that those adaptive transcriptome changes are a consequence of the enrichment of the cell population gene pool with adaptive mutations. Nevertheless, very few studies confronted the transcriptome changes occurring in the cell population during the adaptive phase of continuous culture with the genomic mutations accumulated in the cell population gene pools [8,12].

Since adaptation is the response to the selection pressure of any kind it is tacitly assumed that no changes should occur in cells grown continuously under constant, nonlimiting conditions with no selection pressure applied to them. Yet is it true? This study aimed to get an insight into this matter. In our experimental approach, wild-type W303 strain was grown continuously under stable, nonselective conditions. The stability of culture parameters was assured by a constant flow of nonlimiting growth medium and by maintaining a constant cell density equivalent to the late exponential growth phase. Intriguingly, the transcriptomic analysis of the cell populations revealed over four hundred genes whose expression changed during the steady-state phase of continuous culture. Moreover, these changes were not the consequence of adaptive mutations. Functional analysis of genes whose expression was affected suggested the gradual shift in metabolic balance, the reduction of biosynthetic overcapacity, and the decrease in stress alertness indicating an adaptation to stable, nonstressful growth conditions.

## 2. Materials and Methods

### 2.1. Strain and Media

The W303 (*MAT*a *leu2-3*,*112 trp1-1 can1-100 ura3-1 ade2-1 his3-11*,*15*) wild-type *S. cerevisiae* strain was used in this study. The strain subjected to chemostat experiments was grown in a synthetic minimal liquid medium containing 0.67% yeast nitrogen base (YNB), 2% glycerol, 0.1% yeast extract, and 0.1% glucose and supplemented with the required amino acids and nucleotides. Ampicillin and streptomycin (25 μg/mL) were added to prevent bacterial contamination. For strain propagation and precultures YPD medium (1% Bacto-yeast extract, 2% Bacto-peptone, and 2% glucose with 2% agar for solid medium) was used.

### 2.2. Continuous Culture Conditions

Continuous cultures were conducted in homemade chemostats based on [16]. Briefly, the chambers, containing 20 mL of a synthetic minimal medium, were constructed of 50 mL plastic Falcon tubes each closed with a silicone stopper pierced with three needles of different lengths. The shortest needle was used to add medium to the chamber. The longest needle, which reached the bottom of the tube, provided filtered air (pumped with an aquarium pump) and allowed efficient mixing of cultures. The third needle was used to determine the culture volume by removing effluent from the chamber to the collection of bottles. To start the experiment, 15 mL of liquid YPD medium was inoculated with a single colony and grown overnight at 30 °C. An aliquot of the overnight culture was inoculated to OD600 = 0.1 into a chemostat chamber filled with 20 mL of synthetic minimal medium. When the cell density reached approximately 5 × 10^7^ cells per mL, the flow of medium was turned on at a dilution rate of 0.17–0.18 volumes per hour, and the culture was continued for at least 240 generations at a constant temperature of 30 °C. This dilution rate was selected as optimal for our study in the course of trial yeast cell culture runs. Under these growth conditions, the cell culture density was maintained in a stable state of 5.3 ± 0.3 × 10^7^ cells per mL, the density equivalent to the late exponential phase of batch culture in the same medium. To ensure the stability of the growth parameters, cell culture density was monitored every day, and the medium flow rate was slightly adjusted as necessary. Cultures were also tested for the absence of contamination. These conditions assured constant, unperturbed growth of the cells. The selection of synthetic minimal medium with glycerol as a respiratory carbon source increased the demand for the cells’ metabolic capacity, hence maximizing the number of genes that were expressed in them. Moreover, the lack of selection pressure helped to avoid the gradual accumulation and enrichment of adaptive mutations in the cultured cell populations and bias our results towards adaptation to that pressure. Three such growth experiments were performed comprising independent biological replicates. Cell culture samples (5 mL) were harvested at two time points: after 40 generations (7 days), assumed arbitrarily as the early stage of the steady-state phase, and after 240 generations (42 days), as the adaptation phase. Yeast cells were spun down, flash-frozen in liquid nitrogen, and stored at −80 °C until use.

### 2.3. Preparation of the Total RNA and Microarray Analysis

Total RNA was prepared from deeply frozen yeast cell pellet samples using the hot acidic phenol procedure [17]. The quantity and quality of the RNA preparations were tested with the 2100 Bioanalyzer expert assay RNA Nano (Agilent Technologies, Santa Clara, CA, USA). To identify changes that accumulated throughout the adaptive phase of the continuous culture, comparative microarray hybridizations were performed using the samples after 40 and after 240 generations. Each comparison of the transcriptomes was performed in three biological replicates, each with two technical replicates with a dye-swap. Yeast (v2) Gene Expression 8 × 15 K Microarray slides containing oligonucleotides representing almost all known *S. cerevisiae* genes identified in the SGD were used (Agilent Technologies). Cy3 or Cy5 fluorescently labeled cRNA probes were synthesized using the Agilent Quick Amp Two-Color Labeling Kit according to the manufacturer’s protocol, with total RNA samples as the templates. The probes were hybridized concurrently to microarrays, and the resulting fluorescence images were scanned with the Axon GenePix 4000B (Molecular Devices, San Jose, CA, USA) microarray scanner. Feature extraction was accomplished with GenePix Pro 6.1. Raw data were normalized, averaged, and subjected to statistical analysis with Acuity 4.0 software. Additional data manipulations were performed in Microsoft Excel. For every gene, the value of LogRatio expressing over or underrepresentation of its transcript in cells grown for 240 generations relative to 40 generations was calculated. A gene was considered to be differentially expressed between the evolved and original strains if its transcript level differed between the two strains by at least 1.87-fold (LogRatio ≥ 0.9), and the probability of such a difference by chance was less than 0.05 (*p* < 0.05). The resulting lists of differentially expressed genes (DEGs) (see also microarray analysis results in Appendix A) were subjected to further bioinformatics analysis. Complete transcriptome analysis data are deposited in Gene Expression Omnibus (https://www.ncbi.nlm.nih.gov/geo/) under accession number: GSE162203.

### 2.4. Bioinformatics Analysis

To classify the functions of the identified DEGs, Gene Ontology (GO) analysis was conducted using Cytoscape (v3.8.0) [18] with the BiNGO (v3.0.3) plugin [19] and GO Slim Term Mapper (https://www.yeastgenome.org/goSlimMapper). The raw BiNGO output was edited by removing redundancies in GO terms to improve the clarity of the final list of functional categories overrepresented within the selected gene sets.

To identify transcription factors (TFs) that potentially regulate the expression of selected genes, the Yeastract+ [20] (http://www.yeastract.com) online database was used. We used the tool “Rank Genes by TF” with the filter “Documented Regulations by DNA binding plus expression evidence”. Two queries were performed separately for upregulated and downregulated genes. The resulting lists contained almost 200 TFs, and, to make the list manageable, it was filtered by the statistical significance of this analysis (*p*-value, set arbitrarily to be below 0.005) and by the Fraction number (set to be above 0.06 or 0.08 for up- or downregulated gene sets, respectively) calculated according to this equation:(1)GS(TF)GT(TF)
as the proportion of genes in our set regulated by a given TF (GS(TF)) to the total number of genes in the genome regulated by that TF (GT(TF)). In addition, we calculated the Overrepresentation parameter as the ratio of the number of genes regulated by a given TF in our gene set (GS(TF)) to the total number of genes in this set (GS) divided by the ratio of the number of genes regulated by a given TF in the whole genome (GT(TF)) to the total number of genes in the genome (GT):(2)GS(TF)GSGT(TF)GT

The resulting number is another measure of how much more the particular TF is involved in the regulation of genes from our lists relative to the whole genome. Both Fraction and Overrepresentation parameters detect similar events, but Fraction prefers TFs that appear in the Yeastract database as regulating many, sometimes most, *S. cerevisiae* genes, whereas the Overrepresentation parameter favors TFs that regulate a lower number of genes. The TF lists were further filtered by the Gene count parameter (GS) by removing TFs that regulate fewer than 10 genes. Finally, we added the Rank parameter, which is simply a result of multiplication of Fraction, Gene count, and Overrepresentation.

### 2.5. Quantitative Real-Time RT-PCR Analyses

To validate the microarray results, qRT-PCR was performed with a Mic Real-Time Cycler (BioMolecular Systems, Upper Coomera, QLD, Australia) with fluorescent DNA-binding dye using a QUANTUM EvaGreen^®^ PCR Kit (Syngen Biotech, Warsaw, Poland) according to the manufacturer’s instructions. The primers used to quantify the expression of target genes (Appendix A) were chosen based on data from qPrimerDB (version 1.2). Primer specificity was verified by melting curve analysis. qRT-PCRs were performed in triplicate. Each 20-μL reaction mixture contained 4 μL of QUANTUM EvaGreen^®^ PCR Kit mix (Syngen Biotech, Warsaw, Poland), two primers (4 pmol of each) and 1 μL of template cDNA, which was synthesized from 240 ng of total RNA treated with TURBO™ DNase (Thermo Fisher Scientific, Warsaw, Poland) using a smART First Strand cDNA Synthesis kit (EurX, Gdańsk, Poland) according to the supplier’s protocol. qRT-PCRs were carried out under the following conditions: 95 °C for 15 min, followed by 40 cycles of 95 °C for 15 s, 51.5 °C for 20 s, and 72 °C for 20 s. The crossing threshold cycles (Cq) were determined using micPCR software v2.8.10. Then, the fold changes of the gene expression levels, corrected by efficiency, were analyzed. The *ACT1* gene was used as the normalization reference (internal control) for target gene expression ratios. The Pfaffl method [21] was applied to calculate relative expression with respect to that of *ACT1*.

## 3. Results

Three independent continuous culture experiments were performed for the wild-type haploid W303 strain. In each of these biological replicates, we analyzed transcriptome changes that occurred between the 40th generation time point, i.e., the early stage of the adaptive phase, and the time point after an additional 200 generations of growth in the adaptive phase. The changes in transcriptomes during this growth period were determined by the concurrent hybridization of Cy3- or Cy5-labeled RNA samples to yeast v.2 microarrays. Microarray data were validated by qRT-PCR for 8 randomly selected genes (see Materials and Methods and Appendix A for more details). The correlation coefficient between the microarray and qRT-PCR data for the selected genes was 0.931 (see Appendix A).

Some of the transcriptome changes identified in W303 strain cultures were specific to the individual biological replicates and could be ascribed to the specific mutations found in the gene pools of the respective evolved cell populations. Briefly, in Replicate 1 we detected significant mutations in *HOG1* and *MSY1* genes, in Replicate 2 we found mutations in *HOG1* and *URE2* genes, and in Replicate 3 we identified a mutation in the *MSY1* gene. These data are summarized in the Appendix A. Intriguingly, however, we also noticed changes in the transcriptome that were uniform across biological replicates even though no mutations were affecting the same genes in all three independent cultures of the W303 strain. While these changes were most likely adaptive, they could not have been the consequence of adaptive mutations. The steadiness of the continuous culture conditions ruled out the possibility that the observed transcriptome changes constituted a response to the changing environment. This striking finding encouraged us to investigate further the observed phenomenon.

The criteria for the selection of genes whose expression was uniformly and significantly changed between the two time points were as follows: (|Mean Log_2_Ratio|) ≥ 0.9 AND *p*-value ≤ 0.05 Standard Deviation ≤ 0.8 AND (Standard Deviation)/(Mean Log_2_Ratio)2 ≤ 0.4, where |Mean Log_2_Ratio| is the modulus of the logarithm of fold difference of expression between the two time points averaged across the biological replicates and Standard Deviation is calculated based on the values of these biological replicates. These criteria allowed us to reveal 212 genes whose expression was higher after 240 generations of adaptive growth relative to 40 generations of growth and 241 genes whose expression was lower when comparing the same time points.

The analysis of Gene Ontology terms overrepresentation, performed with Cytoscape (v3.8.0) [18] and the BiNGO (v3.0.3) plugin [19] allowed us to identify several functional categories that were overrepresented within the selected gene sets. The results of this analysis are summarized in Figure 1. Figure 2 shows all the genes in our gene set sorted by their assigned functions. Some categories were merged to reduce their number and increase clarity. Below the gray bars are genes whose functional categories were not overrepresented and genes of unknown function. Genes most strongly up- or downregulated, with LogRatio above 2.33 or below −2.33, respectively i.e., whose expression changed at least 5-fold, are marked in bold face. Appendix A shows the expanded version of Figure 2, listing genes together with their short descriptions and the LogRatio values denoting the increase or decrease of gene expression between 40 and 240 generations.

The data presented allow for several general remarks. Downregulated genes dominated in our gene set. More of them have known functions and more of them belonged to overrepresented Gene Ontology categories compared to upregulated genes. Notably, in both up- and downregulated gene lists there was a 1.5-fold excess of genes qualified in SGD as Uncharacterized relative to the whole *S. cerevisiae* genome (62 of 453 in our gene set versus 578 of 6256 *S. cerevisiae* genes represented by oligonucleotide probes on Yeast (v2) Microarrays). This may have no significance, or it may reveal the existence of novel, unknown and specific functions important for yeast cell survival in continuous culture. These genes may not be well characterized for the very reason that such growth conditions are less frequently employed in studies of yeast cell biology.

### 3.1. Functional Categories of Genes Upregulated during the Prolonged Continuous Culture

Among upregulated genes whose function fell into one of the significantly overrepresented categories, a subset of genes encoding peroxisomal enzymatic proteins involved in the catabolism of fatty acids was recognized. However, we did not observe a significant change in the expression of *PEX* genes involved in peroxisome biogenesis. This indicates that the volume and/or the number of peroxisomes in yeast cells did not change during prolonged culture. Remarkably, the *POT1* gene encoding thiolase performing the last step of the β-oxidation pathway, the release of acetyl-CoA, was not upregulated. Instead, there was an induction of the *ECI1* gene encoding delta3, delta2-enoyl-CoA isomerase. Moreover, the *CIT2* gene encoding peroxisomal citrate synthase that utilizes the β-oxidation pathway end product was downregulated. This expression pattern is compatible with the idea that the role of peroxisomal metabolism under applied growth conditions is not the assimilation of fatty acids but perhaps the remodeling of lipids by introducing double bonds into the fatty acids and thereby increasing membrane fluidity.

Another prominent group contains genes whose products are involved in the catabolism of amines, mostly amino acids, transport, and catabolism of allantoin and other amides. We also identified a group of genes for mitochondrial proteins, both enzymes performing catabolic reactions and mitoribosomal proteins. There is also a subgroup of genes encoding proteins involved in sister chromatid segregation. The purine nucleotide salvage process is represented by two genes, *HPT1* and *XPT1*.

Among the genes of known but not significantly overrepresented functional categories (Figure 2 and Appendix A) there were genes whose products are involved in signal transduction, response to toxins, cell division, metabolism-mostly mitochondrial ion transport and organization, vesicular trafficking, DNA repair, regulation of transcription and translation, RNA metabolism and protein modification. Three groups of genes encoding mitochondrial proteins, chromatin-associated proteins and plasma membrane-associated transporters have functions related to those listed under overrepresented GO categories.

The most strongly upregulated genes encode proteins involved in the transport and catabolism of allantoin (*DAL1*, *DAL4*, *DAL5*); transporter proteins for oligopeptides (*OPT2*), urea and polyamines (*DUR3*); α mating pheromone (*MF(α)2*); proline oxidase (*PUT1*); a transcriptional repressor involved in glucose repression (*MIG2*) and a protein kinase involved in the control of Msn2p-dependent transcription of stress-responsive genes (*YGK3*). 60 upregulated genes were of unknown function.

### 3.2. Functional Categories of Genes Downregulated during Prolonged Continuous Culture

Among downregulated genes whose function fell into one of the significantly overrepresented categories (see Figure 1 and Figure 2 and Appendix A), there was a subset of genes involved in cell wall organization; in the biosynthesis of certain amino acids (note that some other genes for synthesis of other amino acids were upregulated); plasma membrane transporters for ammonium, amino acids (very strongly repressed) and other compounds; sizable groups of genes encoding ribosomal proteins and stress response genes and a large, heterogeneous group of genes involved in the biosynthesis of various compounds.

Among the genes of non-overrepresented functional categories, there were genes encoding proteins localized to various cellular compartments (mitochondria, the cell periphery, endomembrane system, or nuclear pore complex); involved in various aspects of cellular functions (respiration, lipid metabolism, genome maintenance, RNA metabolism, protein modification, invasive growth, and zinc ion homeostasis). There was also an *ATG36* gene required for pexophagy, and downregulation of this gene is compatible with the upregulation of peroxisomal metabolism genes.

Among the most strongly downregulated were genes encoding transporters of amino acids (*BAP3*, *TAT1*, *TAT2*, *AGP1*, *MUP1*, *GNP1*), a protein involved in the regulation of mitochondrial expression of ATP synthase subunits (*NCA3*), ammonium permeases (*MEP1*, *MEP3*), desiccation stress-protecting hydrophilin (*GRE1*), sodium pumps (*ENA1*, *ENA2*), nitroreductase involved in the repression of fatty acid biosynthesis (*FRM2*), glycerol transporter (*STL1*), and several genes of unknown function. Forty-seven downregulated genes had unknown or poorly characterized functions.

Interestingly, there were groups of genes that are functionally related yet were present either in upregulated or in downregulated sets. For instance, genes whose expression was most strongly affected encoded plasma membrane transporters for nutrients. Among those strongly induced were *OPT2*, *DAL5*, *DAL4*, and *DUR3*, whereas *GNP1*, *TAT2*, *MUP1*, *AGP1*, *TAT1*, *MEP3*, *MEP1*, and *BAP3* were strongly downregulated. The distinction seems to be based on ligand specificity: transporters of allantoin, oligopeptides, urea, and polyamines were induced, whereas transporters of amino acids and ammonia were repressed. One could interpret these changes as resulting from a change in the medium composition; however, this was kept constant throughout the entire culture period.

Similarly, while we observed the downregulation of stress-induced and/or stress-related genes (*PUN1*, *YJL144W*, *ECM5*, *DAK2*, *GAD1*, *YPT53*, *TPS2*, *GPP2*, *SNO4*, *HSP32*, *TPS3*, *GPP1*, *TSL1*, *SIP18*, *HSP33*, *EDC2*, *MTL1*, *HSP150*, *DOG2*, *SSA3*, *CIN5*, *PAI3*, *HSP12*, *ALD3*, *CTT1*, *SED1*, *DDR2*, *GRE2*, *STL1*, *FRM2*, *GRE1*), a small number of such genes (*YGK3*, *OYE3*, *ALD2*) were upregulated.

### 3.3. Dubious ORFs

ORFs of this category in most, if not all, cases do not encode real proteins, but microarray probes assigned to these ORFs will detect noncoding RNAs if they are synthesized on the templates of the corresponding genome regions. In many cases, transcript-level information regarding such ORFs may be useful. It is worth noting that the dubious ORF *YGR190C* overlaps with the *HIP1* gene on the opposite strand encoding high-affinity histidine permease [22]. While *YGR190C* is strongly upregulated *HIP1* is on our list of downregulated genes. This suggests the mode of regulation of that gene with the antisense transcription. Such phenomena are quite common [23] and the HIP1 gene may be regulated in a similar fashion, which would explain why *HIP1* and *YGR190C* are inversely regulated.

### 3.4. Search for Common Attributes Affecting the Expression of Selected Genes

To further explore the phenomena occurring within budding yeast cell populations during prolonged continuous culture and the importance of the proteins encoded by genes identified in our transcriptome analysis under these conditions, we looked for the common regulatory systems controlling them.

With the help of the Yeastract+ web tool and database [20] (http://www.yeastract.com) we performed a search for transcription factors (TFs) regulating the expression of the genes on our list. The results of this analysis are shown in Table 1 and Appendix A. The detailed strategy of this search is described in the Materials and Methods section. In brief, we were interested in TFs that regulate many genes in our gene sets, that regulate them preferentially, and for which the result of Yeastract+ analysis was statistically significant. The resulting lists contained 11 TFs for upregulated genes and 25 TFs for downregulated genes. The functions of these TFs, shown in the Description column, generally correlate well with the functions of genes on our lists.

### 3.5. Are There Other Explanations for the Observed Changes in Gene Expression?

Among the strongly upregulated genes were four identical ones, *ASP3-1*, *ASP3-2*, *ASP3-3*, and *ASP3-4*, encoding L-asparaginase. Their increased expression was consistent with the upregulation of other catabolic enzymes; however, all four genes are located in the genome in the immediate proximity of the rDNA region on chromosome XII and also near the Ty1-1 transposon. Mobile elements have been reported to undergo multiplication in the yeast cell genome during long-term continuous culture, with the rate of transposition between 3 × 10^−7^ and 1 × 10^−5^ transpositions per Ty element per generation [24]. The number of rDNA repeats could also have changed throughout our experiment because it is regulated by TOR signaling, with the number of rDNA repeats inversely correlated with the level of the Pnc1p protein [25]. Notably, *PNC1* was downregulated in our list of genes after long-term culture. Moreover, long-term cultivation under selective pressure may favor fitness-increasing duplications of certain regions of chromosomes [12,26]. For all these reasons, we explored the possibility that the upregulation of genes in our study was a consequence of their multiplication, i.e., an increase in transcript level due to an increase in gene dosage even though no selective pressure was applied to cultured cells in our experiments. However, duplication encompasses regions of chromosomes containing at least several genes and would affect the expression of all of them, yet, except *ASP3*, we did not observe the coordinated upregulation of clusters of neighboring genes in our transcriptome analysis results. Thus, while it cannot be absolutely excluded that an increase in *ASP3-1*, *ASP3-2*, *ASP3-3*, and *ASP3-4* expression is the result of their duplication, the change in expression levels of most or all other upregulated genes in our set by their duplication is unlikely.

## 4. Discussion

Since its inception, continuous culture has been used in microevolution studies [27]. More recently, the availability of high-throughput techniques has allowed us to study microevolution by following the adaptive changes of the transcriptomes (e.g., [7,15,28]), proteomes (e.g., [29,30]), metabolomes (e.g., [31]), or genomes (e.g., [9,10,32]) of cell populations subject to continuous culture. It has been assumed that transcriptomic, proteomic, and metabolomic alterations reflect changes in the genomes of the microorganisms under study. In many cases, such an association has been directly proven (e.g., [8,12]) confirming the generally accepted tenet that DNA is a carrier of heritable traits. The observation that became the topic of this work attracted our attention because it implied something different than is commonly believed. Our results revealed that slow, adaptive transcriptome changes in budding yeast cells subjected to microevolution may not always be the consequence of adaptive mutations of their genomes. The expression of over 400 genes was changed between the time point after 40 generations of growth (the beginning of the steady-state phase) and the time point after 240 generations of growth (the adapted cell populations). As there were no common mutations that could be associated with these changes, they must have been encoded in some other way.

The stability of continuous culture conditions in microevolution experiments is usually assured by growth limitation; the medium contains a limiting concentration of one of the nutrients, most often the carbon or nitrogen source [1]. Limitation assures the control of doubling time but at the same time results in starvation stress, which we intended to avoid in our experimental design. The dilution rate was kept at 0.17–0.18 × h^−1^, allowing the cell population to be maintained at a constant density equivalent to the late exponential growth phase. Moreover, to include as many genes as possible in our analysis, we used mineral, YNB-based growth medium with glycerol as a respiratory carbon source (see Materials and Methods).

As documented by Figure 1 and Figure 2, and Appendix A, the modulation of the *S. cerevisiae* cell transcriptome in the course of continuous culture involves groups of genes belonging to distinct functional categories. Evidently, the observed changes in expression levels are specific and signify a shift in the biosynthesis/catabolism balance towards catabolism and towards diminishing the stress response. However, sometimes the cellular processes underpinning these changes are unclear. The observed transcriptome remodeling is interesting in several aspects. First, one might expect a change in gene expression in response to some variations in cell internal or external stimuli of any kind. However, we see the changes despite the absence of such variations. Moreover, changes in gene expression in response to stimuli are fast, measured in minutes or even in seconds, whereas the changes observed by us were slow, measured in days. To our knowledge, this phenomenon has not previously been reported in the literature.

Interestingly, the observed transcriptome remodeling had a negligible effect on the fitness of the cell population. The growth rate of the culture, not limited by the dilution rate, remained constant for the whole duration of culture, even though among differentially expressed genes were those involved in basic cellular functions, such as genes encoding ribosomal proteins. While the observed coordinated transcriptome changes were definitely not random and most likely reflected the cell adaptive response to continuous culture conditions, a much longer continuous culture would probably be necessary to reveal the benefits of these changes.

### 4.1. Regulatory Network Responsible for the Cellular Switch from a Variable to a Stable Environment

More information was extracted by searching for regulatory proteins that may affect the expression of genes in our gene set. This analysis was performed using the Yeastract+ online tool [20]. As demonstrated in Table 1 and Appendix A, there is a very high correlation between the functions of genes in our gene set and the regulatory responsibilities of TFs on the Yeastract+ output list. Within the group of TFs for upregulated genes, we observed those involved in the regulation of catabolism of various nonfermentable carbon sources (Adr1p, Cat8p) and nitrogen sources (Gzf3p, Dal80p, Dal81p). Within the group ascribed to downregulated genes, we observed numerous TFs related to the stress response (Com2p, Crz1p, Hot1p, Rpi1p, Sdd4p, Sfl1p, Sko1p, Wtm2p). We also observed TFs responsible for the induction of filamentous growth (Mig1p, Nrg1p, Nrg2p, Sfl1p), cell wall integrity (Gat4p, Rlm1p, Rpi1p), and membrane rigidity (Mga2p). The association of downregulated genes with the cell periphery, especially with lipid membrane composition, is consistent with the observed upregulation of a subset of genes encoding peroxisomal proteins (see above). TFs responsible for the regulation of amino acid biosynthesis and assimilation (Met4p, Stp1p, Stp2p) were also present. In accordance with the downregulation of ribosomal protein genes, Ifh1p, a TF regulating the transcription of these genes, was present.

Superficially, no clear evidence of the existence of master regulators emerged from this analysis; however, there was a clear tendency toward diminished stress alertness and reduced nutrient reserves and protein biosynthetic capacity. The involvement of TFs, such as Sfl1p and Rpi1p, in more than one of these processes, suggests their coordinated action during adaptation to a less challenging environment. Notably, certain TFs, e.g., Mga2p and Rlm1p, have quite specific regulatory roles in yeast cells according to SGD, yet in the Yeastract+ database, they influence the expression of numerous genes. These or other TFs of still poorly defined regulatory roles may function as switches functioning in the transition from turbulent to tranquil environments. It is also possible that such a switch is of a different nature.

### 4.2. How Could a Constant Cell Environment Induce a Drift in Gene Expression Patterns, and Why Would It Make Biological Sense for the Drift to Be Slow?

The observed changes, downregulation of genes involved in ribosome biogenesis, a shift in balance towards catabolism, and downregulation of stress response and protection genes, suggests that prolonged continuous culture under stable, nonstressful conditions requires fewer resources. This in turn lowers the demand for transactions on the cell periphery, which is exactly what we see.

*S. cerevisiae* cells can occupy various niches differing in nutrient availability and the overall variability of environmental conditions. It is conceivable that survival in unpredictable environments requires a surplus of resources to allow for quick responses to rapid changes. The existence of such overcapacity has been previously reported in microorganisms [33,34], and it has been considered a trait beneficial for fitness under varying growth conditions [35]. It appears that overcapacity is a more general strategy of yeast cell survival in dynamic environments, affecting various cellular functions, such as environmental stress protection, protein biosynthesis, or the ability to acquire and assimilate diverse nutrients [36]. Conversely, in stable conditions, withdrawing overcapacity would be advantageous. Indeed, as a part of adaptive changes occurring in stable chemostat culture conditions, a reduction in this overcapacity was observed for metabolism [14,15,31] and for the signaling networks that regulate it [10]. Therefore, our results are consistent with these observations with one important distinction. Unlike the previous reports mentioned above, our data indicate that adaptation to a stable environment may not involve the accumulation of mutations, so it can be quickly reversed if the conditions become variable again. On the other hand, it seems rational that the changes increasing cell fitness in a stable, nonstressful environment must be slow because the cell must spend some time in such an environment before it is assured that the growth conditions are indeed stable and nonstressful. Such apparent purposefulness is explicable considering the stochastic nature of gene expression. It has been shown that within a genetically homogenous yeast cell population, certain genes, especially those responding to environmental changes, display high cell-to-cell variation in expression levels, termed transcriptional noise [37]. This phenomenon of heterogeneity of gene expression was later shown to be important for survival in conditions of variable nutrient availability [38,39] or stressful environments [40,41]. In addition to genes involved in nutrient assimilation or stress resistance, transcriptional noise is characteristic of genes encoding plasma membrane transporters [42]. Moreover, it is known that stress awareness can be inherited and that the memory of the response to environmental stresses is encoded epigenetically [43,44]. It can be transmitted to offspring as chromatin modifications [45], such as tethering it to the nuclear envelope [46], by histone methylation [47] leading to heterochromatin formation [48], and by changing the nucleosome code [49]. It can also be transmitted by prion propagation [50,51]. These memory records are subject to gradual modifications as the growth conditions change [51]; therefore, the prolonged growth of transcriptionally heterogeneous yeast cell populations in stable, nonstressful conditions will favor cells with lower expression of stress protecting genes whose products are becoming dispensable. This may result in a gradual decrease in their expression levels averaged across the cell population. We did not reveal the specific mechanism(s) pertinent to gene expression differences discovered in our study. However, we postulate that the differences observed by us during the cultivation of *S. cerevisiae* cells in a stable, unchallenging milieu demonstrated the process of “forgetting” their previous experience of growth in variable and stressful conditions.

Incidentally, mutational changes in microevolution experiments leading to the abandonment of network signaling circuits that are dispensable in a constant environment have been considered a manifestation of the myopic nature of evolution; these mutations put cells at a disadvantage when the environment becomes variable again [10]. Our data indicate that switching to a less resource-hungry lifestyle can occur without adaptive mutations and therefore is not so myopic because it can be easily reversed.

To our knowledge, the phenomenon described in this paper has never been the subject of a standalone study; nevertheless, it was recently noticed by others in the course of a large-scale proteomic study [30]. The authors of that paper provide evidence that the gradual changes in protein levels as an adaptation to continuous culture conditions are not a consequence of adaptive mutations. Considering the gradual nature of adaptive changes, the authors suggest the epigenetic mechanism of this regulation [30] and emphasize the presence of a “Histone modifier” GO category among their proteins.

## 5. Conclusions

Slow changes in gene expression in cells grown under stable conditions may be a part of a more general, not yet fully understood, phenomenon. In the majority of studies employing budding yeast or other microorganisms, including those exploring the cell transcriptome or proteome, strains are grown in batch culture and preferably harvested during the exponential growth phase. Less frequently, other growth phases are chosen, and cells are seldom grown for several days or weeks. Therefore, is not surprising that the phenomena specific to these other growth phases have been noticed relatively recently. Interestingly, upregulation of genes encoding peroxisomal proteins together with downregulation of stress-response genes, similar to changes in the transcriptome documented by us, has been observed during prolonged growth of yeast cell colonies [52]. Perhaps cells dwelling in such a microcosm perceive it as stable and nonstressful, which leads to similar transcriptome rearrangements. The observation of slow changes in gene expression in cells grown under constant conditions for many generations reveals gaps in our knowledge of this infrequently studied type of budding yeast lifestyle. At the same time, our results call for greater caution in the interpretation of transcriptome data derived from microevolution experiments.

## Figures and Tables

**Figure 1 genes-11-01419-f001:**
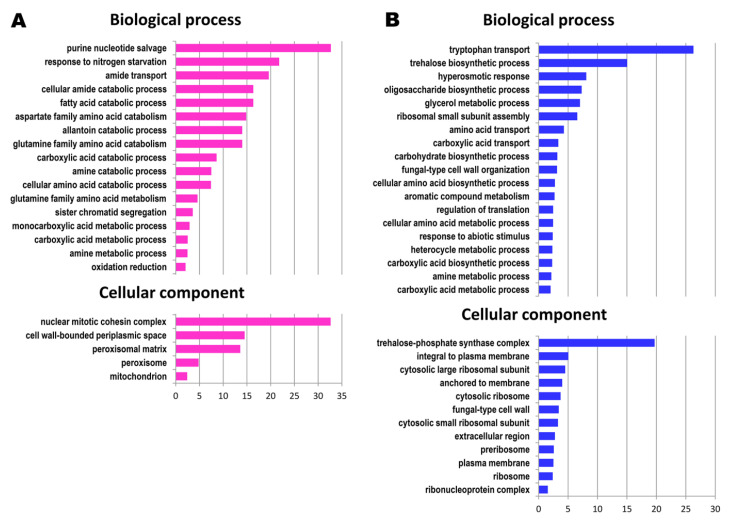
Overrepresentation of Gene Ontology (GO) terms in the group of upregulated (**A**) and downregulated (**B**) genes selected in the course of transcriptome analysis. The analysis was performed using Cytoscape (v3.8.0) [18] with the BiNGO (v3.0.3) plugin [19] and GO Slim Term Mapper (https://www.yeastgenome.org/goSlimMapper). GO terms belonging to Biological process and Cellular Component categories are shown.

**Figure 2 genes-11-01419-f002:**
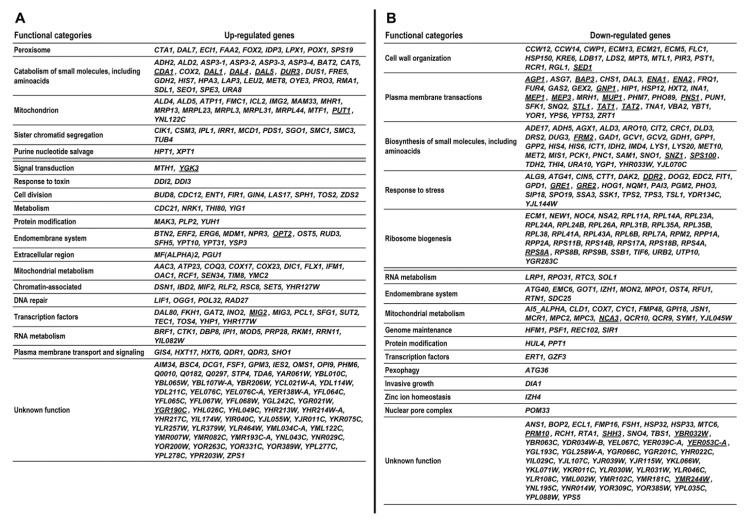
All upregulated (**A**) and downregulated (**B**) genes selected in the course of transcriptome analysis sorted according to the functional category assigned to them. For clarity, the number of GO terms, shown in Figure 1, was reduced by merging the lists of genes of similar functions. Sorting was performed manually based on GO annotations assigned to genes and the descriptions of their functions in the Saccharomyces Genome Database (SGD). Below the gray bar are genes whose functions were not overrepresented within the gene sets. Genes with LogRatio above 2.33 or below −2.33, i.e., whose expression changed at least 5-fold, are marked in bold face. See Appendix A for the expanded version of this figure listing genes together with the short descriptions of their functions taken from SGD.

**Table 1 genes-11-01419-t001:** Transcriptional factors (TFs) are most strongly involved in the regulation of genes selected in transcriptome analysis. The list was generated with the help of the Yeastract+ online database [20] (http://www.yeastract.com). Each TF name is accompanied by the description of the function according to SGD and with the parameters that were used for filtering the raw TF output list. See the Materials and Methods section for a more detailed description of these parameters and the filtering strategy. The lists are sorted according to the Rank parameter.

**Transcriptional Factors for Up-Regulated Genes**						
**TF**	**Fraction**	***p*** **-Value**	**Gene Count**	**Over-represen-tation**	**Rank**	**Description**
Adr1p	0.07	0	54	2.96	11.1	Carbon source-responsive zinc-finger transcription factor; required for transcription of the glucose-repressed gene ADH2, of peroxisomal protein genes, and of genes required for ethanol, glycerol, and fatty acid utilization
Mga1p	0.06	0	54	2.86	9.4	Protein similar to heat shock transcription factor; multicopy suppressor of pseudohyphal growth defects of ammonium permease mutants
Fzf1p	0.09	0	18	4.31	6.8	Transcription factor involved in sulfite metabolism; sole identified regulatory target is SSU1; overexpression suppresses sulfite-sensitivity of many unrelated mutants due to hyperactivation of SSU1
Gzf3p	0.07	0	31	2.51	5.1	GATA zinc finger protein; negatively regulates nitrogen catabolic gene expression by competing with Gat1p for GATA site binding; function requires a repressive carbon source; dimerizes with Dal80p and binds to Tor1p
Dal81p	0.06	0.0001	24	3.12	4.8	Positive regulator of genes in multiple nitrogen degradation pathways
Cat8p	0.09	0	16	3.16	4.6	Zinc cluster transcriptional activator; necessary for derepression of a variety of genes under non-fermentative growth conditions, active after diauxic shift, binds carbon source responsive elements
Dal80p	0.08	0.0003	15	3.18	3.6	Negative regulator of genes in multiple nitrogen degradation pathways; expression is regulated by nitrogen levels and by Gln3p; member of the GATA-binding family, forms homodimers and heterodimers with Gzf3p
Tod6p	0.1	0.0002	10	3.06	3.1	PAC motif binding protein involved in rRNA and ribosome biogenesis; subunit of the RPD3L histone deacetylase complex; Myb-like HTH transcription factor
Sut2p	0.07	0.0012	13	2.35	2.1	Zn2Cys6 family transcription factor; positively regulates sterol uptake under anaerobic conditions with SUT1; represses filamentation-inducing genes during non-starvation conditions; positively regulates mating
Wtm2p	0.07	0.0014	12	2.39	2	Transcriptional modulator; involved in regulation of meiosis, silencing, and expression of RNR genes; involved in response to replication stress
Rpi1p	0.06	0.0053	11	2.13	1.5	Transcription factor, allelic differences between S288C and Sigma1278b; mediates fermentation stress tolerance by modulating cell wall integrity
**Transcriptional Factors for Down-Regulated Genes**						
**TF**	**Fraction**	***p*** **-Value**	**Gene Count**	**Over-represen-tation**	**Rank**	**Description**
Hot1p	0.31	0	32	12.68	126	Transcription factor for glycerol biosynthetic genes; required for the transient induction of glycerol biosynthetic genes GPD1 and GPP2 in response to high osmolarity; targets Hog1p to osmostress responsive promoters
Crz1p	0.11	0	102	3.63	40	Transcription factor, activates transcription of stress response genes; nuclear localization is positively regulated by calcineurin-mediated dephosphorylation
Gis1p	0.12	0	73	3.7	33.5	Histone demethylase and transcription factor; regulates genes during nutrient limitation
Rlm1p	0.11	0	77	3.51	29.4	MADS-box transcription factor; component of the protein kinase C-mediated MAP kinase pathway involved in the maintenance of cell integrity
Ifh1p	0.1	0	44	4.53	20.7	Coactivator, regulates transcription of ribosomal protein (RP) genes
Met4p	0.1	0	63	3.04	18.7	Leucine-zipper transcriptional activator; responsible for regulation of sulfur amino acid pathway
Mig1p	0.1	0	57	3.26	18.2	Transcription factor involved in glucose repression; regulates filamentous growth along with Mig2p in response to glucose depletion
Mga2p	0.08	0	90	2.41	17.6	ER membrane protein involved in regulation of OLE1 transcription; inactive ER form dimerizes and one subunit is then activated by ubiquitin/proteasome-dependent processing followed by nuclear targeting
Stp2p	0.09	0	52	3.33	16.1	Transcription factor; activated by proteolytic processing in response to signals from the SPS sensor system for external amino acids; activates transcription of amino acid permease genes
Stp1p	0.08	0	70	2.62	14.8	Transcription factor; undergoes proteolytic processing by SPS sensor component Ssy5p in response to extracellular amino acids; activates transcription of amino acid permease genes and may have a role in tRNA processing
Nrg1p	0.09	0	46	3.29	14	Transcriptional repressor; mediates glucose repression and negatively regulates a variety of processes including filamentous growth and alkaline pH response
Gzf3p	0.1	0	46	3.06	13.8	GATA zinc finger protein; negatively regulates nitrogen catabolic gene expression by competing with Gat1p for GATA site binding; function requires a repressive carbon source; dimerizes with Dal80p and binds to Tor1p
Sko1p	0.08	0	77	2.12	13.6	Basic leucine zipper transcription factor of the ATF/CREB family; cytosolic and nuclear protein involved in osmotic and oxidative stress responses
Gat4p	0.1	0	42	3.06	13.3	Protein containing GATA family zinc finger motifs; involved in spore wall assembly
Nrg2p	0.16	0	18	4.63	13	Transcriptional repressor; mediates glucose repression and negatively regulates filamentous growth
Hap1p	0.09	0	56	2.52	13	Zinc finger transcription factor; involved in the complex regulation of gene expression in response to levels of heme and oxygen
Rgm1p	0.08	0	49	2.67	10.9	Putative zinc finger transcription factor; overproduction impairs cell growth and induces expression of genes involved in monosaccharide catabolism and aldehyde metabolism; regulates expression of subtelomeric genes
Sfl1p	0.08	0	51	2.53	10.5	Transcriptional repressor and activator; involved in repression of flocculation-related genes, and activation of stress responsive genes; has direct role in INO1 transcriptional memory
Tbs1p	0.13	0	13	3.77	6.2	Putative transcription factor of unknown function
Rof1p	0.09	0	24	2.74	6.1	Putative transcription factor containing a WOPR domain
Tog1p	0.1	0	21	2.93	5.9	Transcriptional activator of oleate genes; regulates genes involved in fatty acid utilization
Wtm2p	0.1	0	17	2.98	5.1	Transcriptional modulator; involved in regulation of meiosis, silencing, and expression of RNR genes; involved in response to replication stress
Rpi1p	0.1	0	17	2.89	4.8	Transcription factor, allelic differences between S288C and Sigma1278b; mediates fermentation stress tolerance by modulating cell wall integrity
Com2p	0.09	0.0001	16	2.94	4.3	Transcription factor; *COM2* transcription is regulated by Haa1p, Sok2p and Zap1p transcriptional activators; C. albicans homolog (MNL1) plays a role in adaptation to stress
Sdd4p	0.1	0.0002	12	2.93	3.5	Putative transcription factor, induced in response to the DNA-damaging agent MMS

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
