# Peer review of "Slow Adaptive Response of Budding Yeast Cells to Stable Conditions of Continuous Culture Can Occur without Genome Modifications"

_genes, 2020, doi:10.3390/genes11121419_

Round 1
Reviewer 1 Report
Please, check some minor details:
-
Lines 93 and 128: S. cerevisiae should be written in italic.
-
Lines 108 and 113: 107 cells - use an uppercase for 7 degree.
-
Line 190: "its derivatives deleted for certain genes" - did you mean "derivatives with deletions of certain genes"? Please, correct.
-
Line 312: "nonoverrepresented" - it is worth writing "non-overrepresented".
In general I suggest to shorten the part of "Results" on 188-259 lines.
Author Response
Reviewer 1
Comments and Suggestions for Authors
Please, check some minor details:
Lines 93 and 128: S. cerevisiae should be written in italic.
Lines 108 and 113: 107 cells - use an uppercase for 7 degree.
Line 190: "its derivatives deleted for certain genes" - did you mean "derivatives with deletions of certain genes"? Please, correct.
Line 312: "nonoverrepresented" - it is worth writing "non-overrepresented".
In general I suggest to shorten the part of "Results" on 188-259 lines.
Response to Reviewer 1
Thank you very much for your comments and suggestions. All mistakes were corrected.
As suggested, the part of the Results section between 188 and 259 lines was shortened. Some text was moved to Supplementary file, some to Materials and Methods section and some was removed altogether.
Reviewer 2 Report
The manuscript by Klim et al., "Slow adaptive response of budding yeast cells to stable conditions of continuous culture can occur without genome modifications" presents an analysis of a transcriptomics dataset obtained from yeast grown in a stable environment of a continuous culture over the course of 200+ generations. From the methodological perspective the work is sound and conducted rigorously, the statistical and enrichment analyses are correct and the discussion of the obtained results is appropriate. The manuscript is well-written and quite pleasant to read.
However, I fail to see the originality, significance and impact of the results presented in the work, as they stand. Chemostat cultures of microorganisms such as yeasts are well known and have been described many times over the past 2 decades, both in studies of adaptive changes and experimental evolution and in investigations of genetic drift in stress-less conditions. The lack of genomic data and knowledge of mutations that occurred in each population prevents proper assessment of the results of the transcriptomics analyses. Granted, the authors openly state that these results are included in a different manuscript, but I think that slicing their data like that truly does a disservice to this otherwise very interesting project and the data should be presented in tandem, otherwise the transcriptomic analyses lack any strong footing.
Author Response
Reviewer 2
Comments and Suggestions for Authors
The manuscript by Klim et al., "Slow adaptive response of budding yeast cells to stable conditions of continuous culture can occur without genome modifications" presents an analysis of a transcriptomics dataset obtained from yeast grown in a stable environment of a continuous culture over the course of 200+ generations. From the methodological perspective the work is sound and conducted rigorously, the statistical and enrichment analyses are correct and the discussion of the obtained results is appropriate. The manuscript is well-written and quite pleasant to read.
However, I fail to see the originality, significance and impact of the results presented in the work, as they stand. Chemostat cultures of microorganisms such as yeasts are well known and have been described many times over the past 2 decades, both in studies of adaptive changes and experimental evolution and in investigations of genetic drift in stress-less conditions. The lack of genomic data and knowledge of mutations that occurred in each population prevents proper assessment of the results of the transcriptomics analyses. Granted, the authors openly state that these results are included in a different manuscript, but I think that slicing their data like that truly does a disservice to this otherwise very interesting project and the data should be presented in tandem, otherwise the transcriptomic analyses lack any strong footing.
Response to Reviewer 2
Thank you very much for your comments and suggestions. The findings reported in this manuscript were made in parallel during the microevolution project that had different objectives and the overall message of the manuscript resulting from that project is quite different. Therefore combining them into one paper would result in work lacking clarity and homogeneity. On the other hand we did not include in this manuscript the details of the findings from the microevolution project to avoid plagiarism. Besides, it would obscure the main message as well as compromise the clarity and homogeneity of this manuscript as well. Nevertheless, the observation we made was so appealing, the reproducibility of transcriptome changes so high that we decided it deserves a separate manuscript. And, as stated in the manuscript, apart from one notice in one publication (cited as [30] in the manuscript), the described phenomenon was, to our knowledge, never reported. Thus, we insist on originality of presented data. Regarding significance and impact, as a person performing transcriptome analyses, therefore watching closely published transcriptome results and methodologies, I see all too often the flaws in those experiments that may result in artifacts. The reason we decided to reveal the existence of the phenomenon described in our manuscript is because it is yet another potential source of artifacts and factors that may skew the whole transcriptome results. We believe that this should be known.
In response to your comment we added to the Supplementary data file a brief summary of the goals and main results reported in the other, microevolution manuscript. We also added the information on the mutations identified in all biological replicates of the wild type W303 strain cell populations after 240 generation of growth. And in the appropriate places of the main text we added reference to this information. Additionally, supplementary Table S3 now includes also the data for genes, for which the change in expression level between 40th and 240th generations was different among biological replicates.
Round 2
Reviewer 2 Report
The revised manuscript addresses my concerns to an acceptable extent, thank you to the authors for providing the extra information.